# Computation of Eigenvalues and Eigenfunctions in the Solution of Eddy Current Problems

**DOI:** 10.3390/s23063055

**Published:** 2023-03-12

**Authors:** Theodoros Theodoulidis, Anastassios Skarlatos, Grzegorz Tytko

**Affiliations:** 1Department of Mechanical Engineering, Faculty of Engineering, University of Western Macedonia, ZEP Campus, 50150 Kozani, Greece; 2Commissariat à l’Énergie Atomique et aux Énergies Alternatives (CEA), Laboratory for Integration of Systems and Technology (LIST), Université Paris-Saclay, F-91120 Palaiseau, France; 3Faculty of Automatic Control, Electronics and Computer Science, Silesian University of Technology, Akademicka 16, 44-100 Gliwice, Poland

**Keywords:** nondestructive testing, eddy current testing, eigenvalues and eigenfunctions, complex roots

## Abstract

The solution of the eigenvalue problem in bounded domains with planar and cylindrical stratification is a necessary preliminary task for the construction of modal solutions to canonical problems with discontinuities. The computation of the complex eigenvalue spectrum must be very accurate since losing or misplacing one of the thereto linked modes will have an important impact on the field solution. The approach followed in a number of previous works is to construct the corresponding transcendental equation and locate its roots in the complex plane using the Newton–Raphson method or Cauchy-integral-based techniques. Nevertheless, this approach is cumbersome, and its numerical stability decreases dramatically with the number of layers. An alternative, approach consists in the numerical evaluation of the matrix eigenvalues for the weak formulation for the respective 1D Sturm–Liouville problem using linear algebra tools. An arbitrary number of layers can thus be easily and robustly treated, with continuous material gradients being a limiting case. Although this approach is often used in high frequency studies involving wave propagation, this is the first time that has been used for the induction problem arising in an eddy current inspection situation. The developed method is implemented in Matlab and is used to deal with the following problems: magnetic material with a hole, a magnetic cylinder, and a magnetic ring. In all the conducted tests, the results are obtained in a very short time, without missing a single eigenvalue.

## 1. Introduction

Heng to the increase in computer processing power, mathematical models have become an integral part of the comprehensively conducted eddy current testing. Such models are utilized at each test stage, starting from the designing of eddy current probes, through the selection of optimal test parameters and carrying out simulations, to the interpretation of the obtained results and their implementation directly in the measuring device. Several eddy current problems have been solved with the application of extremely fast and effective analytical models. A significant reduction in computing time is obtained through abandoning the modeling of the infinite domain, which has made possible to replace infinite integrals and series with a finite number of terms. This procedure is computationally efficient and errors are easily controlled by simply adjusting the location of the truncation boundaries or by modifying the number of terms in the eigenfunction expansion. A truncated domain, however, implies a description of the field by discrete eigenfunction expansions and eigenvalues.

In canonical geometries eigenvalues are computed as roots of expressions involving trigonometric or Bessel functions. In the case of modeling objects with large geometric dimensions, such as a plate [1,2,3,4,5] or half-space [6,7], the region under consideration consists exclusively of conductive material. In this case, the eigenvalues are real numbers, and their calculation is relatively easy. What poses a real challenge is the determination of complex eigenvalues when the region under consideration consists of several sub-regions (containing conductive material or air). Such a situation occurs when modeling disks [8,9,10,11], tubes [12,13,14,15,16], rods [17,18], materials with a defect [19,20,21,22,23,24], and wherever there are edges [25,26,27,28,29] or discontinuities [30,31,32] (Figure 1). Determination of the eigenvalues then boils down to finding complex roots of the appropriate complex function.

Because of the oscillating nature of eigenfunctions, and overflow errors caused by local maxima of very large values, both complex plane search methods and the procedures available in numerical packages (such as fsolve in Matlab or FindRoot in Mathematica) are often not fully efficient at precisely determining all eigenvalues. At the same time, attempts to develop a reliable algorithm based on iterative methods, such as the Newton–Raphson method, have been unsuccessful so far, even with the employment of a very small step. More promising seems to be the approach based on Cauchy’s argument principle [33,34,35,36,37,38,39,40] the solution domain is divided into small parts where the roots are found with contour integration. Unfortunately, this is a time-consuming method that requires carrying out numerous integrations, and the roots located too close to the contour edge are omitted anyway.

The lack of a fast and reliable method for finding complex eigenvalues significantly hinders the creation of new analytical models, and consequently restricts the development of eddy current modeling. What is particularly notable is the lack of a universal root-finding algorithm. Therefore, each new eddy current problem requires introducing necessary modifications to the applied solution. In an attempt to meet these requirements, this paper proposes a completely different and much more efficient approach, wherein the same algorithm is used for each configuration. Starting from the Helmholtz equation for the magnetic vector potential, we treat a general Sturm–Liouville problem and apply linear algebra theory to transform the problem of finding the complex roots to that of finding the eigenvalues of a matrix.

The developed solution can be used for a wide class of eddy current problem, in particular for those related to the detection of flaws and discontinuities in tested objects. Its speed and efficiency have been verified with the use of other root-finding algorithms, with very good results. The proposed algorithm turns out to be the only one in the entire set of tests that does not omit a single eigenvalue, while performing the calculations in a very short time.

## 2. Theory

The solution to eddy current problems can be derived from the Helmholtz equation for the magnetic vector potential **A**, which for constant or piecewise constant magnetic permeability *μ_r_* and harmonic excitation of frequency *f* takes the form:(1)$$\nabla^{2}A+k^{2}A=0,$$
where *k*^2^ = −*j ω μ*_0_
*μ_r_ σ,* with *ω =* 2*πf* denoting the angular frequency and *σ* the conductivity. This equation can be solved either as three scalar Helmholtz equations in the Cartesian coordinate system or it can be further scalarized by using the second-order vector potential defined by
(2)A=∇×W.

In all cases, the resulting equations are independent scalar Helmholtz equations of the form:(3)$$\nabla^{2}U(x,y,z)+k^{2}U(x,y,z)=0.$$

The solution to (3) using separation of variables is generally possible for constant *k* and simple geometries involving infinite planes and cylindrical configurations (layered or not). When either *k* is a continuous function of a coordinate variable or the geometry involves edges or discontinuities, a possible solution is the use of the truncated region eigenfunction expansion (TREE) method [26]. The method involves the truncation of the solution domain in a suitable direction depending on the problem geometry. The diffusive nature of an eddy current problem implies that the truncation can be done without introducing significant errors, provided that the truncation boundaries are sufficiently far such that the field is small. The most significant advantage of the approach is the ability to match interface conditions across several boundaries simultaneously and, hence, to obtain solutions for new configurations by adopting traditional canonical structures.

The air-core coil positioned over the conductive material with a hole of radius *c* is shown in a cylindrical coordinate system (Figure 2). After application of the method of separation of variables we end up with the following differential equation for the *r*-dependence of the magnetic vector potential in magnetic truncated domains with a conductivity that varies with *r* [41].
(4)$$\frac{\partial^{2}A}{\partial r^{2}}+\frac{1}{r}\frac{\partial A}{\partial r}-\frac{A}{r^{2}}-k^{2}(r)A+\mu_{r}\frac{\partial (1/\mu_{r})}{\partial r}\cdot \frac{1}{r}\frac{\partial (rA)}{\partial r}=0,$$
where *k*^2^(*r*) = *j ω μ*_0_ *μ_r_ σ*(*r*).

For a truncated domain, assuming a Dirichlet boundary condition at *r* = *h*, constant conductivity, and constant magnetic permeability, the eigenvalues *u_m_* can be obtained from the roots of the Bessel function of the first kind:(5)$$J_{1}(u_{m}h)=0.$$

The eigenvalues of (4) in the general case of a varying *k*^2^ and *μ_r_* can be found by applying Sturm–Liouville theory. The Sturm–Liouville problem consists in finding eigenvalues and eigenfunctions for the following general differential equation, which is the *r*-dependence of *U* in the method of separation of variables:(6)$$-\frac{\partial }{\partial r}\left(P(r)\frac{\partial X(r)}{\partial r}\right)+R\left(r\right)X(r)=\lambda W\left(r\right)X(r).$$

The generalized eigenvalues problem transforms into a matrix eigenvalues calculation [42]:(7)AX=λBX,
where
(8)$$A_{mn}=\int_{0}^{h}\left[P(r)\frac{d\varphi_{m}(r)}{dr}\frac{d\varphi_{n}(r)}{dr}+R(r)\varphi_{m}(r)\varphi_{n}(r)\right]dr,$$
(9)$$B_{mn}=\int_{0}^{h}W(r)\varphi_{m}(r)\varphi_{n}(r)dr,$$
and it has been assumed that a solution to (3) has the form:(10)$$X_{m}(r)=\sum_{n}V_{mn}\varphi_{n}(r).$$

The basic functions *ϕ_n_*(*r*) are the solutions of a simpler differential equation and satisfy the boundary conditions at *r* = 0, *h*. In our case, from (5).

## 3. Solution

The application of the developed solution for the computation of complex eigenvalues is presented for three canonical eddy current problems. Material with a hole, the conductive cylinder, and the magnetic ring are considered as axisymmetric problems and examined in a cylindrical coordinate system. In the case of conductive material with a hole (Figure 2), (4) can be written as:(11)$$-\frac{\partial^{2}A}{\partial r^{2}}-\frac{1}{r}\frac{\partial A}{\partial r}+\frac{A}{r^{2}}+k^{2}(r)A-\mu_{r}\frac{\partial (1/\mu_{r})}{\partial r}\cdot \frac{1}{r}\frac{\partial (rA)}{\partial r}=\lambda A,$$
then converted to the form:(12)$$-\frac{\partial }{\partial r}\left[\frac{1}{r}\frac{\partial (rA)}{\partial r}\right]+k^{2}(r)A-\mu_{r}\frac{\partial (1/\mu_{r})}{\partial r}\cdot \frac{1}{r}\frac{\partial (rA)}{\partial r}=\lambda A,$$
and finally written as:(13)$$-\mu_{r}r\frac{\partial }{\partial r}\left[\frac{1}{\mu_{r}}\frac{1}{r}\frac{\partial (rA)}{\partial r}\right]+\frac{k^{2}(r)}{\mu_{r}r}(rA)=\lambda (rA).$$

We deduce the following coefficients by comparing (13) to (6):(14)$$W(r)=\frac{1}{\mu_{r}r},$$
(15)$$P(r)=\frac{1}{\mu_{r}r},$$
(16)$$R(r)=\frac{k^{2}(r)}{\mu_{r}r},$$
(17)X(r)=rA.

Taking into account the Dirichlet boundary conditions, at *r* = 0, *h* the basic functions are *ϕ_n_*(*r*) = *r J*_1_(*u_n_ r*), with *u_n_* given by (5). The coefficient *k*^2^(*r*) has the form:(18)$$k^{2}(r)=\{\begin{array}{cc} 0 & 0\leq r\leq c\\ j\omega \mu_{0}\mu_{r}\sigma & c\leq r\leq h, \end{array}$$
and the matrices (8) and (9) take the analytical form:(19)$$\begin{array}{l} A_{mn}=\frac{1}{\mu_{r}}(u_{m}^{2}+k^{2})\frac{h^{2}}{2}J_{0}^{2}(u_{m}h)I\\ -\frac{1}{\mu_{r}}k^{2}\{\begin{array}{cc} \frac{c}{u_{m}^{2}-u_{n}^{2}}\left[u_{n}J_{0}(u_{n}c)J_{1}(u_{m}c)-u_{m}J_{0}(u_{m}c)J_{1}(u_{n}c)\right] & m\neq n\\ \frac{c^{2}}{2}\left[J_{1}^{2}(u_{m}c)-J_{0}(u_{m}c)J_{1}(u_{m}c)\right] & m=n, \end{array} \end{array}$$
(20)$$B_{mn}=\{\begin{array}{cc} 0 & m\neq n\\ \frac{1}{\mu_{r}}\frac{h^{2}}{2}J_{0}^{2}(u_{m}h)I & m=n. \end{array}$$

Thus, if we solve the matrix eigenvalues problem of (7) with the matrices defined as **A***_mn_* and **B***_mn_*, we have found the sought complex eigenvalues and the corresponding eigenfunctions in the form of eigenvectors. Such a computation is supported by every mathematical software package.

The region containing material with a hole (Figure 2) consists of two sub-regions: air (0 ≤ *r* ≤ *c*) and conductor (*c* ≤ *r* ≤ *h*). Through changing the order of the sub-regions, i.e., the conductor (0 ≤ *r* ≤ *c*) and air (*c* ≤ *r* ≤ *h*), a conductive cylinder with radius *c* is obtained. In practical applications, this type of configuration is often used for modeling tested objects, such as rods, disks, pucks or coins (Figure 3). By adopting a procedure analogous to that used for the material with a hole, a solution for a conductive cylinder is obtained.
(21)$$k^{2}(r)=\{\begin{array}{cc} j\omega \mu_{0}\mu_{r}\sigma & 0\leq r\leq c\\ 0 & c\leq r\leq h, \end{array}$$
(22)$$\begin{array}{l} A_{mn}=\frac{1}{\mu_{r}}u_{m}^{2}\frac{h^{2}}{2}J_{0}^{2}(u_{m}h)I\\ +\frac{1}{\mu_{r}}k^{2}\{\begin{array}{cc} \frac{c}{u_{m}^{2}-u_{n}^{2}}\left[u_{n}J_{0}(u_{n}c)J_{1}(u_{m}c)-u_{m}J_{0}(u_{m}c)J_{1}(u_{n}c)\right] & m\neq n\\ \frac{c^{2}}{2}\left[J_{1}^{2}(u_{m}c)-J_{0}(u_{m}c)J_{1}(u_{m}c)\right] & m=n, \end{array} \end{array}$$
(23)$$B_{mn}=\{\begin{array}{cc} 0 & m\neq n\\ \frac{1}{\mu_{r}}\frac{h^{2}}{2}J_{0}^{2}(u_{m}h)I & m=n. \end{array}$$

The proposed method can be used for any number of sub-regions. In the case of the conductive ring (Figure 4), the solution domain contains three sub-regions: air (0 ≤ *r* ≤ *c*_1_), conductive material (*c*_1_ ≤ *r* ≤ *c*_2_), and air (*c*_2_ ≤ *r* ≤ *h*). This geometry may describe, for example, the eddy current nondestructive inspection of a tube. The solution can be written as:(24)$$k^{2}(r)=\{\begin{array}{cc} 0 & 0\leq r\leq c_{1}\\ j\omega \mu_{0}\mu_{r}\sigma & c_{1}\leq r\leq c_{2}\\ 0 & c_{2}\leq r\leq h, \end{array}$$
(25)$$\begin{array}{l} A_{mn}=u_{m}^{2}\frac{h^{2}}{2}J_{0}(u_{m}h)+(\frac{1}{\mu_{r}}-I)u_{m}u_{n}\;[G(c_{2}{)}_{mn}\\ -G(c_{1}{)}_{mn}]+\frac{k^{2}}{\mu_{r}}[F(c_{2}{)}_{mn}-F(c_{1}{)}_{mn}], \end{array}$$
(26)$$B_{mn}=\frac{h^{2}}{2}J_{0}^{2}(u_{m}h)+(\frac{1}{\mu_{r}}-I)[F{\left(c_{2}\right)}_{mn}-F{\left(c_{1}\right)}_{mn}],$$
where
(27)$$G{\left(x\right)}_{mn}=\{\begin{array}{cc} \frac{x}{u_{m}^{2}-u_{n}^{2}}\left[u_{m}J_{0}(u_{m}x)J_{1}(u_{m}x)-u_{n}J_{0}(u_{m}x)J_{1}(u_{n}x)\right] & m\neq n\\ \frac{x^{2}}{2}\left[J_{0}^{2}(u_{m}x)-J_{1}^{2}(u_{m}x)\right] & m=n, \end{array}$$
(28)$$F{\left(x\right)}_{mn}=\{\begin{array}{cc} \frac{x}{u_{m}^{2}-u_{n}^{2}}\left[u_{n}J_{0}(u_{n}x)J_{1}(u_{m}x)-u_{m}J_{0}(u_{m}x)J_{1}(u_{n}x)\right] & m\neq n\\ \frac{x^{2}}{2}\left[J_{1}^{2}(u_{m}x)-J_{0}(u_{m}x)J_{2}(u_{m}x)\right] & m=n. \end{array}$$

## 4. Results and Discussion

The developed solution is named the Sturm–Liouville global function (SLGF) method and is implemented in Matlab. Matrix eigenvalues calculations (7) are make using the command [V,D] = eigs(A,B,‘sr’), with V representing the eigenvectors and D the set of eigenvalues (a matrix whose diagonal elements are equal to the eigenvalues squared). Since accuracy in the numerical computation is higher for the first eigenvalues, for *N_S_* eigenvalues, we use matrices with dimensions 4*N_S_* × 4*N_S_*. The eigenvalues determined in this way are verified with the multilevel computation of complex eigenvalues (MCCE) method [39], the Newton–Raphson method and the fsolve() procedure available in Matlab. The calculations consist in finding the *Ns* = 50 first eigenvalues for the sets of input data (Table 1) that correspond to different values of the parameters used in eddy current testing. The examination of whether a given complex number is a correctly calculated eigenvalue is carried out with the employment of Cauchy’s argument principle, based on the integration of a precisely determined contour. The number of incorrect eigenvalues (missing or false) obtained by each of the methods is presented in Table 2. The times taken to obtain results on a computer with an Intel Core i5 processor and 6 GB of RAM memory, are also compared.

The SLGF method presented in this paper is the only one that makes it possible to find all eigenvalues in each test. The results are obtained in a very short time, which usually does not exceed 1 s. Such high efficiency is ensured due to the transformation of the problem under consideration into the matrix eigenvalues calculation in the form presented in (7). In this way, a significant independence of the calculation process from the values of input parameters is achieved. As for the other methods, the effectiveness of determining eigenvalues depends primarily on the value of the coefficient *k*^2^ = *j ω μ*_0_
*μ_r_ σ*. As the value of *k* increases, numerical errors may appear. This is a direct result of the property of the function that ensures the continuity of the magnetic field for *r* = *c*, and which is used to find eigenvalues. These properties may cause numerous limitations, e.g., difficulties in the determining of eigenvalues for high frequencies (tests 2 and 9).

The great strength of the SLGF method lies in a simple numerical implementation that contains only two full matrices (**A** and **B**). Unlike in the case of other solutions, no sets of initial points or integration operations are used there. What is more, there is no need to create procedures for filtering the resulting set of values, that is, removing either zero roots, or multiple, false or negative sign roots. It is worth noting that, in the developed solution, no algorithm for splitting the domain of the solution is employed to determine the eigenvalues. Such algorithms are utilized in methods based on Cauchy’s argument principle, where the domain within which the searched roots are located is split into parts in the form of contours. This division is usually quite complicated because each contour should contain a few roots at the most. In addition, the roots that are located too close to the contour edge are missed. Together with the increase in coefficient k, the densification of roots increases (the difference in the values of successive roots is smaller and smaller), so the precise determination of contours becomes more and more difficult (tests 2, 3, 5, 9).

In the case of calculating the changes in the impedance of the air-core coil over conductive material with a hole (Figure 2) with the employment of analytical models, finding a set of eigenvalues is by far the most time-consuming process. The calculation of the change in the impedance of such a coil using the TREE method takes about 1 s (tests 1–6), of which 0.6–0.7 s is taken by the process of finding all eigenvalues (Table 2). With the finite element method (FEM), it takes about 10 s to obtain the change in coil impedance. However, the highest reduction in the length of time needed to obtain results, in comparison with the FEM method, can be achieved during calculations performed for many iterations, when precomputations are used in the analytical models, and thus the eigenvalues are determined only once throughout the calculation cycle [43].

## 5. Conclusions

A novel method for computing complex eigenvalues in eddy current problems is presented. It has been applied to the simulation of the air-core coil described above: conductive cylinder, material with a hole and magnetic ring. What is characteristic of the developed solution is simple numerical implementation and versatility. When dealing with other eddy current problems, it is enough to find a new form of matrix **A**, **B** and coefficient *k*, without the necessity to modify the very method. The presented approach can be used for both magnetic and non-magnetic materials containing flaws, discontinuities, and edges. The conducted tests show that the SLGF method is the only one that does not omit any eigenvalue, making it possible to obtain results in a very short time. The changes in the coil impedance calculated on the basis of the determined eigenvalues show an excellent agreement in comparison with the results obtained with the FEM method.

In future work, it is planned to adapt the SLGF method for other eddy current problems, in particular testing of materials consisting of many sub-regions, thermal barrier coatings (TBC), and media with piecewise magnetic permeability. The presented approach will also be utilized for 3D simulations of problems to which the analytical solution has not been worked out yet.

## Figures and Tables

**Figure 1 sensors-23-03055-f001:**
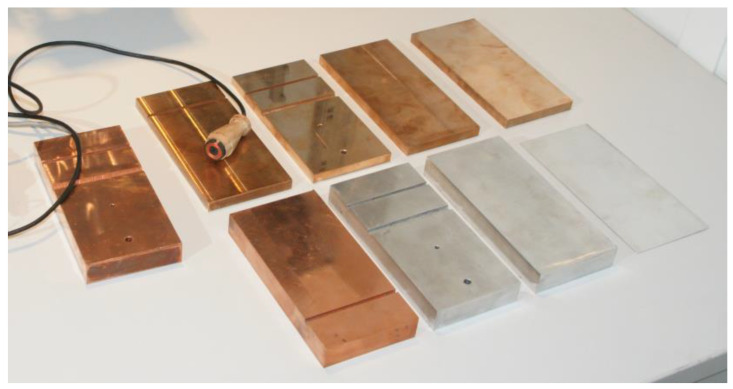
The eddy current probe with samples for testing holes, slots, and edge effects.

**Figure 2 sensors-23-03055-f002:**
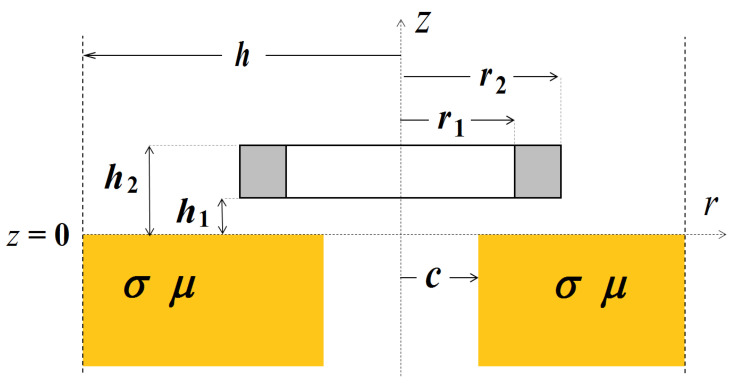
The air-core coil above the conductive material with a hole in a cylindrical coordinate system.

**Figure 3 sensors-23-03055-f003:**
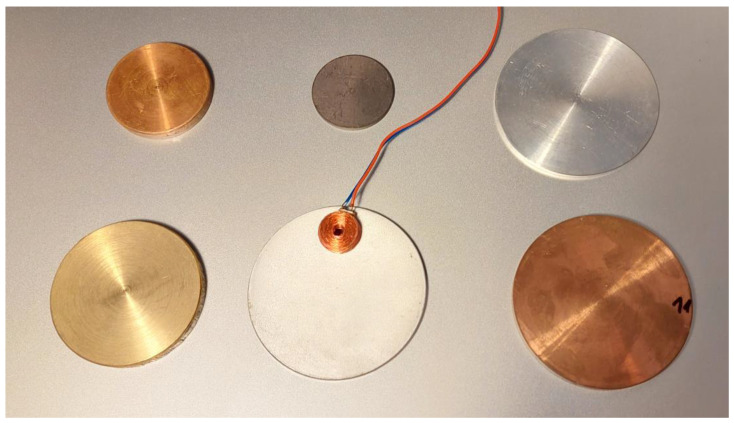
The air-core coil and conductive disks.

**Figure 4 sensors-23-03055-f004:**
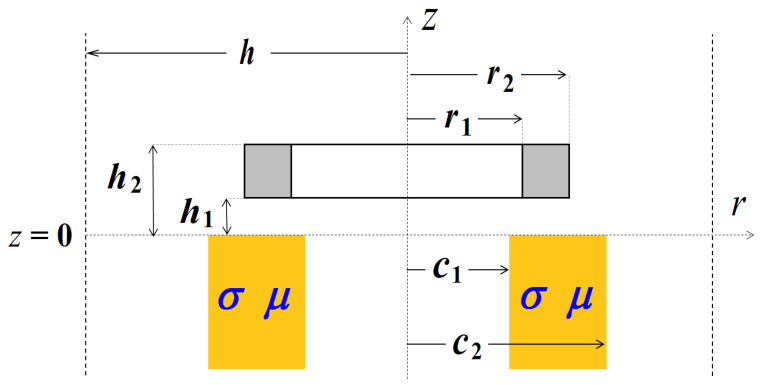
The air-core coil located above the magnetic ring in a cylindrical coordinate system.

**Table 1 sensors-23-03055-t001:** Input data sets used in the calculations.

Tests	*f* [kHz]	*μ_r_*	*σ* [MS/m]	*c* [mm]	Problem
1	1	1	60	4	hole
2	100	1	60	4	hole
3	10	50	60	4	hole
4	1	50	1	4	hole
5	10	10	30	8	hole
6	10	1	30	8	hole
7	10	1	60	15	disk
8	1	10	1	10	disk
9	200	1	30	15	disk
10	10	1	1	5	disk

**Table 2 sensors-23-03055-t002:** Time for computation and number of incorrect eigenvalues.

Tests		Incorrect Eigenvalues				Time [s]		
	SLGF	MCCE	Newton	Fsolve	SLGF	MCCE	Newton	Fsolve
1	0	0	0	0	0.6	1.0	0.4	2.3
2	0	2	3	3	0.7	7.5	17.7	9.8
3	0	3	3	3	0.6	2.0	13.1	1.1
4	0	0	0	0	0.6	1.1	0.4	1.4
5	0	6	6	6	0.6	2.1	12.6	0.9
6	0	0	0	0	0.7	1	7.2	1.1
7	0	6	1	6	0.6	5.4	12.9	5.0
8	0	0	1	0	1.0	2.1	0.2	1.7
9	0	12	12	12	2.0	2.1	11.5	1.7
10	0	0	0	0	0.6	1.1	1.2	6.4

## Data Availability

Not applicable.

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
