# Peer review of "Computation of Eigenvalues and Eigenfunctions in the Solution of Eddy Current Problems"

_sensors, 2023, doi:10.3390/s23063055_

Round 1

Reviewer 1 Report

This maucript proposes a Sturm-Liouville global function (SLGF) method to deal with the solution of the differential equation that describes eddy current probelm.  The developed method was successfully implemented in Matlab and validated with the employment of the Cauchy’s argument principle. The paper is well formated and adequately described, and the novelty and significance of content is high. Based on these considerations, I think this paper can be accepted in the present form.

Author Response

Dear Editor,

We would like to thank the reviewers for a number of constructive comments for improving the readability of the manuscript and for pointing out certain deficiencies. We have made suggested improvements at appropriate places in the revised manuscript (blue text) according to the following replies to the reviewers’ comments.

Reviewer 1

  • This maucript proposes a Sturm-Liouville global function (SLGF) method to deal with the solution of the differential equation that describes eddy current probelm.  The developed method was successfully implemented in Matlab and validated with the employment of the Cauchy’s argument principle. The paper is well formated and adequately described, and the novelty and significance of content is high. Based on these considerations, I think this paper can be accepted in the present form.

We would like to thank you very much for your report, which motivates us to further research.

Reviewer 2 Report

The authors deal with an interesting approach to the design of a method for evaluating the manifestations of eddy currents, for example, used in NDD. For eddy current problems in bounded domains, an important aspect of the solution of the relevant differential equation is the exact calculation of the discrete eigenvalues and their corresponding eigenfunctions.
For the conducting materials currently in use, these eigenvalues are generally complex and are evaluated as roots of expressions involving trigonometric or Bessel functions. So far, the methods used to find and quantify these roots have been complex plane search methods involving Newton-Raphson iterations or the Cauchy integral approach, etc..
In the present study, the authors have sought a solution in the form of an alternative path for the formulated differential equation describing the electromagnetic field in the form of a solution to the general Sturm-Liouville problem. They proposed a global function method to transform the modeled/solved problem into an eigenvalue matrix problem (mathematical model of the problem).
The proposed method with experimental background was developed and implemented in Matlab environment and was used to solve the following tests: geometry and material of magnetic material with hole, magnetic cylinder and magnetic ring. In all the tests performed, results were obtained in a very short time without any major error or fundamental mistake. Not a single eigenvalue of the analyzed problem was missing.
The method and content is interesting and beneficial in the field, and contains elements of novelty. The work is at a good theoretical level and contains experimental validation.
From a formal point of view, I have no major reservations.
No plagiarism was found.

Author Response

Dear Editor,

We would like to thank the reviewers for a number of constructive comments for improving the readability of the manuscript and for pointing out certain deficiencies. We have made suggested improvements at appropriate places in the revised manuscript (blue text) according to the following replies to the reviewers’ comments.

Reviewer 2

The authors deal with an interesting approach to the design of a method for evaluating the manifestations of eddy currents, for example, used in NDD. For eddy current problems in bounded domains, an important aspect of the solution of the relevant differential equation is the exact calculation of the discrete eigenvalues and their corresponding eigenfunctions.
For the conducting materials currently in use, these eigenvalues are generally complex and are evaluated as roots of expressions involving trigonometric or Bessel functions. So far, the methods used to find and quantify these roots have been complex plane search methods involving Newton-Raphson iterations or the Cauchy integral approach, etc..
In the present study, the authors have sought a solution in the form of an alternative path for the formulated differential equation describing the electromagnetic field in the form of a solution to the general Sturm-Liouville problem. They proposed a global function method to transform the modeled/solved problem into an eigenvalue matrix problem (mathematical model of the problem).

The proposed method with experimental background was developed and implemented in Matlab environment and was used to solve the following tests: geometry and material of magnetic material with hole, magnetic cylinder and magnetic ring. In all the tests performed, results were obtained in a very short time without any major error or fundamental mistake. Not a single eigenvalue of the analyzed problem was missing. The method and content is interesting and beneficial in the field, and contains elements of novelty. The work is at a good theoretical level and contains experimental validation. From a formal point of view, I have no major reservations. No plagiarism was found.

Thank you very much for such a detailed review and positive opinion about our work.

Reviewer 3 Report

The similarity report of the article is attached. The abstract part of the article sent for publication with the attached work of the authors is exactly the same. The difference between these studies should be explained.

The paper in its  current form can be improved by considering the following queries:

1. What is really motivation of this study and its novelty and what is the real-life application of the present study

2. The authors might rewrite the section “Abstract”

My opinion: The resuls of this manuscript seem interesting and mathematically correct. In my opinion the manuscript is suitable for publication in your journal after the minor revision mentioned above.

With best regards

Author Response

Dear Editor,

We would like to thank the reviewers for a number of constructive comments for improving the readability of the manuscript and for pointing out certain deficiencies. We have made suggested improvements at appropriate places in the revised manuscript (blue text) according to the following replies to the reviewers’ comments.

Reviewer 3

The similarity report of the article is attached. The abstract part of the article sent for publication with the attached work of the authors is exactly the same. The difference between these studies should be explained.

  1. The authors might rewrite the section “Abstract”

Thank you for this comment. Abstract has been re-written.

“The solution of the eigenvalue problem in bounded domains with planar and cylindrical stratification is a necessary preliminary task for the construction of modal solutions to canonical problems with discontinuities. The computation of the complex eigenvalues spectrum must be very accurate since loosing or misplacing one of the thereto linked modes will have an important impact to the field solution. The followed approach in a number of previous works is to construct the corresponding transcendental equation and locate its roots in the complex plane using the Newton-Raphson method or Cauchy-integral-based techniques. Nevertheless, this approach is cumbersome, and its numerical stability decreases dramatically with the number of layers. An alternative, approach consists in the numerical evaluation of the matrix eigenvalues for the weak formulation for the respective 1D Sturm-Liouville problem using linear algebra tools. An arbitrary number of layers can be thus easily and robustly treated, with continuous material gradients being a limiting case. Although this approach is frequently used in high frequency studies involving wave propagation, it is the first time that is used for the induction problem arising in eddy current inspection situation. The developed method is implemented in Matlab and is used to deal with the following problems: magnetic material with a hole, a magnetic cylinder, and a magnetic ring. In all the carried out tests, the results are obtained in a very short time, without missing a single eigenvalue.”

The paper in its  current form can be improved by considering the following queries:

  1. What is really motivation of this study and its novelty and what is the real-life application of the present study

We have described and explained these issues in the Introduction and Conclusion sections.

“What poses a real challenge is the determination of complex eigenvalues when the region under consideration consists of several sub-regions (containing conductive material or air). Such a situation occurs while modeling disks [8]-[11], tubes [12]-[16], rods [17]-[18], materials with a defect [19]-[24], and wherever there are edges [25]-[29] or discontinuities [30]-[32] (Figure 1).”

“The lack of a fast and reliable method of finding complex eigenvalues significantly hinders the creation of new analytical models, and consequently restricts the development of eddy current modeling. What is particularly notable is the lack of a universal root-finding algorithm. Therefore each new eddy current problem requires introducing necessary modifications to the applied solution. In an attempt to meet these requirements, this paper proposes a completely different and much more efficient approach, wherein the same algorithm is used for each configuration.”

“The developed solution can be used for a wide class of eddy current problems, in particular for those related to the detection of flaws and discontinuities in tested objects.”

“In future work, it is planned to adapt the SLGF method for other eddy current problems, in particular testing of: materials consisting of many sub-regions, thermal barrier coating (TBC), and media with piecewise magnetic permeability. The presented approach will also be utilized for 3D simulations for the problems to which the analytical solution has not been worked out yet.”

My opinion: The resuls of this manuscript seem interesting and mathematically correct. In my opinion the manuscript is suitable for publication in your journal after the minor revision mentioned above.

Thank you very much for all your comments and positive opinion about our manuscript.